# A New Sternorrhynchan Genus and Species from the Triassic Period of China That Is Likely Related to Protopsyllidioid (Insecta, Hemiptera) [note 1]

**DOI:** 10.3390/insects13070592

**Published:** 2022-06-28

**Authors:** Diying Huang, Marina Hakim, Yanzhe Fu, André Nel

**Affiliations:** 1State Key Laboratory of Palaeobiology and Stratigraphy, Center for Excellence in Life and Paleoenvironment, Nanjing Institute of Geology and Palaeontology, Chinese Academy of Sciences, Nanjing 210008, China; marina@nigpas.ac.cn (M.H.); yzfu@nigpas.ac.cn (Y.F.); 2Institut de Systématique, Evolution, Biodiversité (ISYEB), Muséum National d’Histoire Naturelle, CNRS, Sorbonne Université, EPHE, Université des Antilles, CP 50, 57 Rue Cuvier, 75005 Paris, France

**Keywords:** Psyllaeformia, Permopsyllidiidae, phylogeny, Middle Triassic, Yanchang Formation, Tongchuan biota

## Abstract

**Simple Summary:**

The new sternorrhynchan *Triassopsyllidiida pectinata* gen. et sp. nov. is described from the latest Middle Triassic Tongchuan biota of China. The phylogenetic relationships of the Protopsyllidioidea are discussed.

**Abstract:**

*Triassopsyllidiida pectinata* gen. et sp. nov. is described from the latest Middle Triassic Tongchuan biota of China and tentatively attributed to the superfamily Protopsyllidioidea. Its forewing venation is unique among this superfamily in the anteriorly pectinate vein ScP + RA and the presence of a veinlet between R and M + CuA. Its exact position in this group remains uncertain, mainly because of the weak diagnostic value of the wing venation characters in these insects. The phylogenetic relationships of the Protopsyllidioidea are discussed.

## 1. Introduction

The Protopsyllidioidea Carpenter, 1931, is a group of extinct Psyllaeformia with fossils ranging between the Late Permian and the Late Cretaceous, and is currently subdivided into four families: Postopsyllidiidae Hakim, Azar & Huang, 2019; Permopsyllidiidae Becker-Migdisova, 1985; Protopsyllidiidae Carpenter, 1931; and Paraprotopsyllidiidae Hakim, Azar, Szwedo, Drohojowska & Huang, 2021. After the morphological phylogenetic hypothesis of Hakim et al. [1], the Protopsyllidioidea would be a grade that leads to the modern clade (Psylloidea + Aleyrodomorpha). This hypothesis does not fit with the molecular phylogeny of the Hemiptera proposed by Song et al. [2], who obtained a clade (Psylloidea (Aleyrodoidea (Coccoidea + Aphidoidea))), instead of two clades (Psylloidea + Aleyrodoidea) and (Coccoidea + Aphidoidea) as in Hakim et al. [1]. These last authors have neglected the important character presence versus absence of the crossvein cua-cup between the veins CuA and CuP’, an absence shared by all the extant (Psylloidea (Aleyrodoidea (Coccoidea + Aphidoidea))) while it is present in the fossil Protopsyllidioidea (except the Paraprotopsyllidiidae), Archescytinidae, etc. Thus a new morphological analysis is necessary to resolve this contradiction.

Here, we describe a new Triassic fossil that we tentatively attribute to the Protopsyllidioidea. It greatly differs from all the described representatives of this group. As well, it should be helpful for the future phylogenetic analysis of these insects.

## 2. Material and Method

Only one specimen, with a part and counterpart, was preserved in the greenish-grey shale of a lacustrine deposit of the Tongchuan biota. It was collected in the lower parts of the Middle–Upper Triassic Yanchang Formation near Hejiafang Village, Jinsuoguan Township, Yintai District, Tongchuan City, Shaanxi Province, northern China (for detailed location of fossil site, see Fu et al. [3]). This layer is the upper section of Ch7 Member of the formation cited as the Tongchuan Formation in most palaeontological studies. The age of the fossil layers are very close to the Middle–Late Triassic boundary [4] and assigned to the latest Ladinian in this paper.

The specimens were carefully prepared using a sharp blade by removing the rock that was covering part of it. Photographs were taken with a Zeiss AxioZoom V16 stereoscope. All images were optimised and grouped into plates using Adobe Photoshop CS6. Line drawings were drafted with Adobe Illustrator CC 2018. The material studied here is deposited in the Nanjing Institute of Geology and Palaeontology, Chinese Academy of Sciences, Nanjing, China.

We follow the wing venation nomenclature of Nel et al. [5]. Abbreviations: ScA: subcostal anterior; ScP: subcostal posterior; RA: radius anterior; RP: radius posterior; MA: median anterior; MP: median posterior; CuA: cubitus anterior; CuP: cubitus posterior; cua-cup specialized crossvein between CuA and CuP; rp-m distal crossvein between RP and M.

## 3. Results


*Systematic palaeontology*


Order Hemiptera Linnaeus, 1758

Suborder Sternorrhyncha Amyot & Audinet-Serville, 1843

Infraorder Psyllaeformia Verhoeff, 1893

? Superfamily Protopsyllidioidea Carpenter, 1931

? Family Permopsyllidiidae Becker-Migdisova, 1985


***Triassopsyllidiida* gen. nov.**


*Type species*. *Triassopsyllidiida pectinata* sp. nov.

*Etymology*. Named after the Triassic period and the suffix ‘psyllidiid’. The gender is feminine. The genus is registered under urn:lsid:zoobank.org:act:FF62F8E4-C4D4-47A9-AF4A-2438E7F769A6.

*Diagnosis*. Nearly all main veins with strong setae, except CuP; ScP + RA anteriorly pectinate with eight branches; RP simple; no crossvein rp-m; MA simple; MP posteriorly pectinate; a long and narrow areola postica; veins of anal area forming a Y-vein.


***Triassopsyllidiida pectinata* sp. nov.**



**(Figure 1, Figure 2 and Figure 3)**


*Holotype*. NIGP180297 (part and counterpart of fragmental body with a forewing), stored at Nanjing Institute of Geology and Palaeontology, CAS, Nanjing, China.

*Type locality*. Latest Middle Jurassic at the locality near the Hejiafang village, Jinsuoguan Township, Tongchuan City, Shaanxi Province, North China; Late Ladinian, Middle Triassic.

*Etymology*. Named after the pectinate vein ScP + RA. The species is registered under urn:lsid:zoobank.org:act:52254537-208D-44D3-8A63-999730432F3D.

*Description*. Head ovoid-shaped; eyes large, ellipse-shaped; protothorax wider than head; mesothorax wide; legs covered with fine setae, middle legs long and slender, hind legs distinctly longer than middle legs, hind femur robust, tibia much longer than femur (Figure 1). Forewing (Figure 2 and Figure 3), 3.6 mm long, 1.4 mm wide; no trace of coloration visible; ambient vein present but not very thick; all main veins except CuP with large and robust setae regularly spaced in one row, basal part of wing margin armed with one row of small setae, middle and posterior parts of wing margin gradually become armed with setae; C with small setae; ScA long, with large and robust setae, very close to C and reaching C at beginning of pectinate area of RA + ScP; ScP oblique, appressed to stem of R, joining RA at point of separation between RA and RP; a strong convex vein R + M + CuA, M + CuA emerging 0.6 mm from wing base, RP emerging from R 0.4 mm from base of M + CuA, simple and straight, reaching wing apex; ScP + RA slightly curved, pectinated before mid-wing, with seven branches, all simple except third with one secondary branching, most basal one (possibly ScP) being slightly stronger than more distal ones; distal stem of M + CuA 0.5 mm long; a short veinlet between R and M + Cu; stem of M 0.9 mm long, separating into MA and MP; MA simple, reaching wing apex; MP posteriorly regularly pectinate with four branches; stem of CuA 0.5 mm long, with two long branches defining a long and narrow areola postica, CuA2 straight, CuA1 curved; we cannot decipher the presence vs. absence of crossvein cua-cup between CuP and M + CuA, slightly arched, without setae; CuP weakly indicated, concave and straight, declined near its apex; two veins in anal area, distally fused and forming an Y-vein.

## 4. Discussion

Among the Permian–Triassic hemipteran groups, only the sternorrhynchan Protopsyllidioidea have all the following characteristics of *Triassopsyllidiida* gen. nov.: area between C and ScP + R rather narrow; no or very few crossveins, especially in distal half of wing; RP simple; a long stem M + CuA emerging from R + M + CuA; presence of strong setae on all main veins [1].

Drohojowska et al. [6] proposed the following synapomorphies for the Sternorrhyncha: ‘rostrum placed in sternal depression’, ‘mesonotum strongly raised’, ‘mesoscutum forming two humps’, ‘veinlet cua-cup at basal cell absent’. The last character state is erroneous because the crossvein cua-cup is clearly present in the Protopsyllidioidea (see [1]). The body characters are unknown in *Triassopsyllidiida* gen. nov. Drohojowska et al. [6] considered the Protopsyllidioidea as a sister to the clade [[Dinglomorpha + Aleyrodomorpha] + [Liadopsyllidae + Psylloidea]]. This clade is supported by: ‘vein Pc not carinate’, ‘vein ScP, short, not reaching margin’, ‘thickened ambient vein present’, ‘abdomen narrowly connected to thorax’, and ‘hypandrium present’. The ambient vein of *Triassopsyllidiida* gen. nov. is very clear; the shape of Pc vein is unknown. The vein ScP can have different shapes among the Protopsyllidioidea, but generally it does not end in C but in R—as in *Triassopsyllidiida* gen. nov.—supporting its attribution to this group.

Drohojowska et al. [6] did not propose any synapomorphy supporting the Protopsyllidioidea. The venation of *Triassopsyllidiida* gen. nov. is very different from that of the Aleyrodomorpha and Dinglomorpha (Dinglidae Szwedo & Drohojowska, in [6]), especially in the presence of an areola postica. The Psylloidea have a very different radius anterior RA.

Some other fossil families show similarities with *Triassopsyllidiida* gen. nov., namely the Liadopsyllidae Martynov, 1926, but these have no strong setae on their veins and their vein M has only two distal branches, unlike *Triassopsyllidiida* gen. nov. [7,8,9]. The Archescytinidae Tillyard, 1926, and the Pincombeidae Tillyard, 1922, have a simple RP, but their M and CuA separate at their common base on R + M + CuA [10,11]. The Dysmorphoptilidae Handlirsch, 1906, also has a simple RP, or it is weakly forked, and a common stem of M with CuA, but they have a broader area between C and ScP + R and a tegmen that is abruptly narrowed distally. None of these groups have strong macrotrichia on their veins [12,13,14,15].

The Permian genus *Orthoscytina* Tillyard, 1926 (in Prosboloidea, a ‘problematic paraphyletic group’ in Cicadomorpha, after Szwedo et al. [16]), shares with *Triassopsyllidiida* gen. nov. the simple RP, M with several branches, and CuA with at least two branches. Even an unnamed specimen figured by Evans [17] has the very particular anteriorly pectinate ScP + RA of *Triassopsyllidiida* gen. nov., but *Orthoscytina* strongly differs from *Triassopsyllidiida* gen. nov. in the very broad area between C and ScP + R, where ScP and R are basally separated and there is the absence of a stem of M + CuA re-emerging from R + M + CuA.

Hakim et al. [1] proposed the following wing venation characters for the Protopsyllidioidea: forewing with stem R branched, ScP developed variously, in some taxa delimiting pterostigma posteriorly (closed pterostigma), or weakened (opened pterostigma); RA and RP always present; stem M fused with stem CuA at base for at least a short distance, M two-, three-, rarely four-branched; CuA forked, areola postica present [but reduced in some taxa, e.g., *Angustipsyllidium minutum* Hakim, Azar et Huang, 2021], branch CuA2 sometimes weakened; clavus triangular, shortened with two claval veins, or A2 fused with margin. All these characters are present in *Triassopsyllidiida* gen. nov.

Hakim et al. [1] characterized the venation of the Mesozoic Postopsyllidiidae as follows: forewings setose; pterostigma greatly reduced (sometimes pigmented); vein M three-branched; A1 and A2 present and well developed. *Triassopsyllidiida* gen. nov. has a M with five branches, excluding it from this family. Hakim et al. (2019) also gave the following characteristics in their diagnosis of the Protopsyllidiidae: forewing with pterostigma absent (while present in *Triassopsyllidiida* gen. nov., but also in at least the type genus *Protopsyllidium*, see Tillyard (1926)), basal cell normally absent; vein M two-branched (unlike *Triassopsyllidiida* gen. nov., but this difference is more of generic importance than sufficient to discriminate families), vein CuA forked, rarely simple; crossvein rp-m normally absent.

Hakim et al. [1] also proposed the following diagnostic characters for the Permopsyllidiidae: forewings wide and round near apex, narrower at base, no setae on veins or wing margin; when present, vein ScA (?) short; vein R normally branches around mid-length of wing; vein RP arcuate or slightly curved, rarely straight; pterostigma wide, triangular and sometimes darkened; vein M with three or four branches; one or two crossveins rp-m visible; basal cell present in several taxa; anal area wide; veins A1 and A2 connected distally into a Y-shape vein. The Permopsyllidiidae differ from *Triassopsyllidiida* gen. nov. in the absence of setae on veins and the presence of crossvein(s) between RP and M. The absence versus presence of setae on veins is sometimes hard to determine on fossils, as demonstrated by their bases, which are clearly visible on the imprint of the holotype of *Triassopsyllidiida pectinata* sp. nov., while they are not detectable on its counter-imprint. Thus, this character remains uncertain. The presence versus absence of crossvein rp-m can be more diagnostic, at least at the genus level, but it is hard to determine its value at the family level and it is not stable in the family.

The number of branches of M varies in *Permopsyllidium* from three branches in *Permopsyllidium lesclansis* Prokop et al., 2015; *Permopsyllidium mitchelli* Tillyard, 1926; and *Permopsyllidium affine* Tillyard, 1926; to four branches in *Permopsyllidium australense* Becker-Migdisova, 1985 or *Permopsyllidium quadrimediatum* Becker-Migdisova, 1985; but in that case, M is forked twice, while it is pectinate in *Triassopsyllidiida pectinata* sp. nov. [8,18,19].

*Triassopsyllidiida* gen. nov. differs from all the genera currently in the Protopsyllidioidea in the presence of a series of straight, parallel veinlets between ScP + RA in the distal half of forewing, and the presence of a short veinlet between R and M + CuA [1,19,20,21,22,23,24,25,26,27,28,29]. Only *Triassopsylla* Tillyard, 1917, has a ScP re-emerging distally from RA and a distal fork of RA, which is very different from the series of eight veinlets emerging from ScP + RA as in *Triassopsyllidiida* gen. nov. [30]. One would be tempted to create a new family for *Triassopsyllidiida* gen. nov. on the basis of this curious character, but such veinlets distally emerging from ScP + RA are also present in some extant species of Psylloidea, e.g., *Euphyllura pakistanica* Loginova, 1973, while the other representatives of *Euphyllura* have only two branches of ScP + RA [31]. Thus, this character alone is not sufficient to support a family. Nevertheless, *Triassopsyllidiida* gen. nov. strongly differs from all genera of the Protopsyllidioidea. The second character, ‘presence of a short veinlet between R and M + CuA’, is also very unusual among the Protopsyllidioidea, but it would be necessary to find it on further specimens of *Triassopsyllidiida* gen. nov. to determine if it is stable or not before erecting a new family for *Triassopsyllidiida* gen. nov.

After the phylogenetic hypothesis of Hakim et al. [29], *Triassopsyllidiida* gen. nov. would fall in the clade (Postopsyllidiidae (Protopsyllidiidae (Psylloidea + Aleyrodomorpha))) on the basis of the absence of the crossvein rp-m, which is a homoplastic character. These authors did not consider the presence of robust setae on the veins in their analysis, a character present in the Postopsyllidiidae, the Protopsyllidiidae, and the Paraprotopsyllidiidae [1]. It would be necessary to verify the absence of these setae in all the previously described Permopsyllidiidae. These setae are sometimes hardly visible in fossils. The character, ‘presence of strong setae on main veins’, could be a potential synapomorphy for the clade Protopsyllidioidea.

The exact position of *Triassopsyllidiida* gen. nov. remains somewhat uncertain because it does not have the putative apomorphy of the Postopsyllidiidae (absence of pterostigma), and neither that of the Protopsyllidiidae (vein M reduced to two branches). The Paraprotopsyllidiidae have a simple median vein. Hakim et al. [29] did not propose any synapomorphy for the Permopsyllidiidae. *Triassopsyllidiida* gen. nov. does not fit with the diagnoses of the four families Postopsyllidiidae, Protopsyllidiidae, Paraprotopsyllidiidae, and Permopsyllidiidae. The differences in forewing venation between all these families are relatively weak characteristics, rendering their definitions and limits somewhat uncertain, especially for the Permopsyllidiidae that are only based on forewing characters.

## 5. Concluding Remarks

The Permopsyllidiidae are currently restricted to the Upper Permian Period of Australia and the Russian Federation, while the Postopsyllidiidae and Paraprotopsyllidiidae are Cretaceous, and the Protopsyllidiidae are known in the Permian of Australia, France, the Russian Federation, and South Africa; the Triassic of South America, Australia, and South Africa; the Jurassic of China, Europe, and Central Asia; and the Cretaceous on nearly all the paleo-continents. These families seem to be very ancient and were probably very widely distributed during the period between the late Permian and the late Cretaceous. While the Protopsyllidioidea survived the Permian–Triassic episode, known as the ‘mother of all extinctions’, the causes and exact period(s) of their extinctions remain obscure. It is probably not due to competition with some other acercarian groups because they co-habited with the stem groups of the modern sternorrhynchan clades for a long time. A link with the floristic changes of the early Late Cretaceous could also be unlikely, as the youngest representatives are known in the Raritan amber (Turonian). They possibly became extinct during the period between the Turonian and the Maastrichtian, when the ‘Angiosperm Terrestrial Revolution’ was at its maximum [32], or at the very end of the Cretaceous, as for the Odonata Aeschnidiidae, which was very recently described from the latest Maastrichtian [33]. Unfortunately, the current knowledge on the entomofaunas from this crucial period is very fragmentary.

## Figures and Tables

**Figure 1 insects-13-00592-f001:**
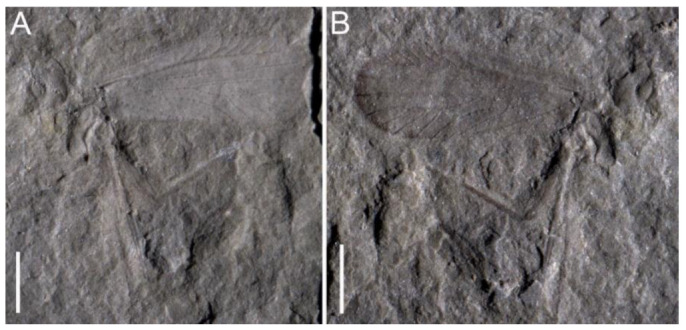
*Triassopsyllidiida pectinata* gen. et sp. nov., Middle Triassic Tongchuan biota, holotype NIGP180297, photographs of general habitus. (**A**) part; (**B**) counterpart. Scale bars: 1 mm.

**Figure 2 insects-13-00592-f002:**
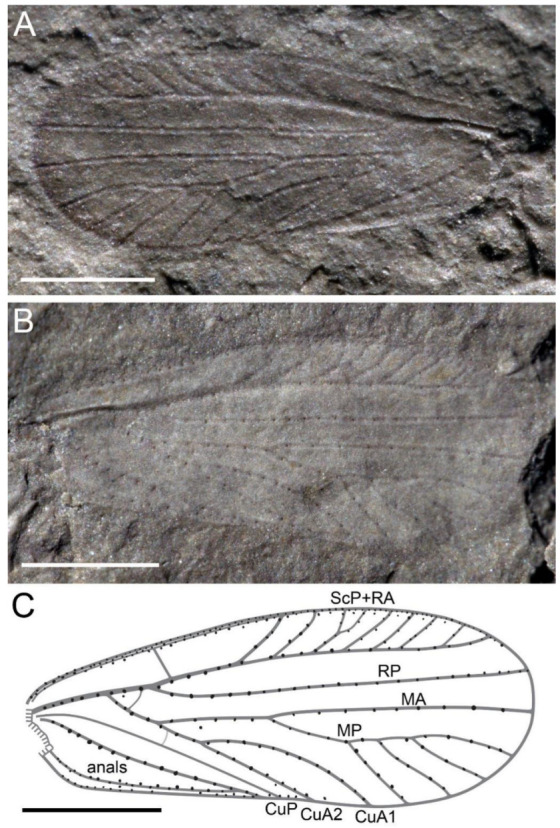
*Triassopsyllidiida pectinata* gen. et sp. nov., Middle Triassic Tongchuan biota, holotype NIGP180297, photographs of wing. (**A**) wing of counterpart; (**B**) wing of part; (**C**) line drawing. Scale bars: 1 mm.

**Figure 3 insects-13-00592-f003:**
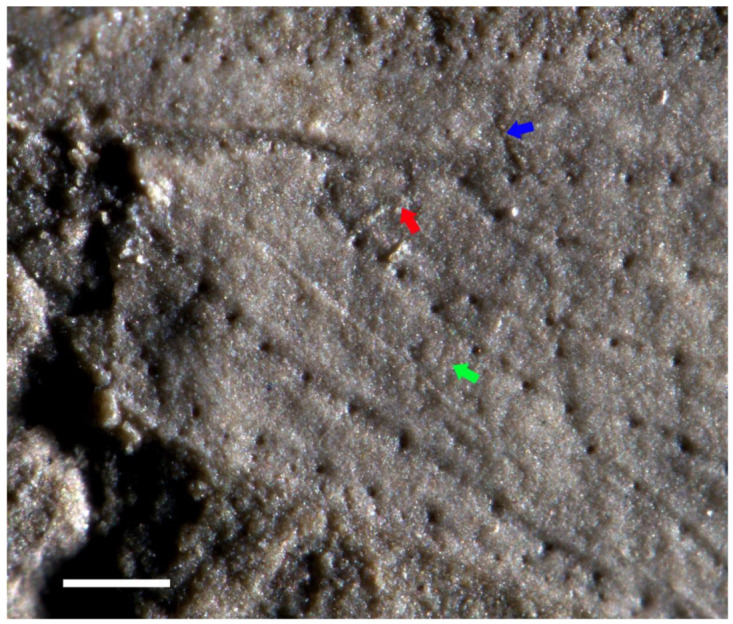
*Triassopsyllidiida pectinata* gen. et sp. nov., Middle Triassic Tongchuan biota, holotype NIGP180297, photographs of wing base, blue arrow indicating a putative crossvein between ScA and ScP+R, red arrow indicating a putative crossvein between R and M + CuA, and green arrow indicating a possible crossvein cua-cup. Scale bars: 0.2 mm.

## Data Availability

The data presented in this study are available in the article.

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
