# Peer review of "A New Sternorrhynchan Genus and Species from the Triassic Period of China That Is Likely Related to Protopsyllidioid (Insecta, Hemiptera)†"

_insects, 2022, doi:10.3390/insects13070592_

Round 1

Author Response

We have followed all the remarks

except for putting in italics the names of crossveins, these are veins, and it is not necessary

Reviewer 2 Report

I think it is a well-written paper to read. Actually I learned lot from this work. Only two minor comments below:

1. Is it possible to mark the vein on drawing (like Fig. 2) as Insects is not a traditional taxonomic journal. Appropriate wing vein illustration may make the article much more reader-friendly.

2. Line 141: a RP is a typo?

Author Response

all changes accepted, thanks a lot

Reviewer 3 Report

It is an interesting paper on a new genus and species of Hemiptera Sternorrhyncha from the Triassic of China

I only made some comments and suggestions on the manuscript.

Page 1

Line 40: improve the wording

Page 2

Line 65: word change suggestion

Line 66: Add Zoobank registration number

Page 4

Line 86: Remove

Line 90: word change suggestion

Line 91: Add Zoobank registration number

Line 92: improve the wording

Page 5

Line 120: Add a comma

Lines 123-128: improve the wording

Line 134: word change suggestion

Line 141: there is a crossed out "a"

Line 146: Add “and”

Page 6

Line 161: there is a crossed out "a"

Line 212: improve the wording

Author Response

all changes accepted and done, thanks a lot